# Optical Properties of LiGdF$_4$ Single Crystal in the Terahertz and Infrared Ranges

Gennady A. Komandin [1,2], Sergey P. Lebedev [1], Stella L. Korableva [3], Oleg A. Morozov [4], Vladimir M. Kyashkin [5], Vadim V. Semashko [4] and Pavel P. Fedorov [1,*]

[1] Prokhorov General Physics Institute of the Russian Academy of Sciences, 38 Vavilov St., Moscow 119991, Russia
[2] Faculty of Laser and Optoelectronic Systems, Bauman Moscow State Technical University, 5 Baumanskaya 2nd St., Moscow 105005, Russia
[3] Institute of Physics, Kazan Federal University, 18 Kremljovskja Str., Kazan 420008, Russia
[4] Zavoisky Physical-Technical Institute, FRC Kazan Scientific Center of RAS, 10 Sibirsky Trakt Str., Kazan 420029, Russia
[5] Institute of High Technologies and New Materials, National Research Ogarev Mordovia State University, 68 Bolshevistskaya Str., Saransk 430005, Russia
* Correspondence: ppfedorov@yandex.ru; Tel.: +7-915-160-36-40

**Abstract:** The basic mechanisms of absorption of THz range radiation in optically perfect LiGdF$_4$ single crystals were studied using the broadband experimental data and the dielectric response function analysis within the harmonic oscillator model. The polarized IR reflection spectra have allowed one to determine the phonon contribution in the absorption coefficient in the THz range, while transmission spectra in the THz range were used to obtain the birefringence value and the effects of various mulitparticle processes. Additionally, we established the optical and electrodynamic parameters of the LiGdF$_4$ single crystal, which are necessary for the design of nonlinear optical devices.

**Keywords:** fluoride single crystals; terahertz and infrared spectroscopy; phonon and multiparticle absorption; dielectric response function



## 1. Introduction

In addition to the obvious coverage of the frequency range of low-energy vibrational, rotational, and relaxation excitations in various media, terahertz (THz) radiation has well-known advantages [1]. The energy of THz radiation quanta is less than meV, and this is not enough for the ionization of matter, in contrast to X-ray (tens of keV) irradiation. The penetration depth of THz waves depends on the type and structural and morphological properties of matter. Scattering effects in porous and heterogeneous substances are noticeably reduced for wavelengths of 100 μm or longer. Materials with such properties are in demand in the introscopy (defectoscopy) of modern materials [2,3], high-rate communication systems, environmental monitoring, and the detection of hazardous substances [4–6]. The safety of THz radiation for biological objects opens up broad prospects for its application in medicine for the diagnosis and treatment of various diseases [7–9].

The increasing efficiency of optical systems in the THz range poses the problem of synthesizing and studying new materials for the manufacture of passive and active devices, such as reflectors, lenses, splitters, waveguides, converters, amplifiers, and generators. Single-crystal fluorine compounds have prospects for application in THz-range systems. In particular, LiLnF$_4$ (Ln = Tb-Yb) crystals are used in broadband optical insulators based on the Faraday effect [10]. The successful use of the LiYF$_4$:Nd laser in a compact THz-radiation generator was reported in [11,12].

In addition to laser physics application [13–15], scheelite-type LiGdF$_4$ crystals (space symmetry group $I4_1/a$, Z = 4 [16]) are used as one of the most efficient magnetic adiabatic

refrigerators for obtaining ultralow temperatures [17]. This may be of interest for the elaboration of highly sensitive systems for detecting THz radiation. Fluoride crystals are widely used as active laser media in the mid-IR to VUV range [18], but their characteristics in the THz spectrum range require further careful study.

The optical and vibrational properties of $LiLnF_4$ ($Ln$ = Y, Gd-Lu) (scheelite crystal family) were studied using IR, Raman, and inelastic neutron scattering data [19–21]. Nevertheless, there are no reports on the optical characteristics of the said crystals in the THz range. Standard approaches to the analysis of the IR spectra, including the generalized four-parameter model and classical oscillator model, provide limited reliability to the results extrapolated to both low and high frequency with respect to phonon eigenfrequencies. In addition to the contribution of the low-frequency wing of phonon absorption, crystal absorption increases in the THz range due to extrinsic and multi-particle processes [22,23]. The high-frequency extrapolation of the calculated spectra of optical phonons, obtained with the use of the classical oscillator model, gives unacceptable overestimated values due to the assumption that the damping constant does not depend on frequency. At the same time, narrow-band measurements by THz spectroscopy only allow one to determine the optical constants of the crystal, which are the sum of various processes. Thus, the calculation of the contributions of all absorption mechanisms requires the acquisition of broadband data covering the entire range dispersion of the analyzed processes.

This paper presents the results of our study of the scheelite-type $LiGdF_4$ single crystal.

## 2. Experimental Methods and Data Analysis

Single crystals $LiGdF_4$ were grown in Kazan Federal University by the Bridgman–Stockbarger method (32 mol% $GdF_3$ and 68 mol% LiF composition; 755 °C; $10^{-3}$ Pa high vacuum; high-purity graphite resistive heater and thermal insulators; 1 mm/h pulling rate; graphite crucibles for [100]-oriented boules [24]). These parameters are consistent with the growing conditions selected by Chai et al. [25].

X-ray diffraction (XRD) measurements were performed on Empyrean X-ray diffractometer (Malvern PANalytical B.V.; Cu K$\alpha$ radiation, $\lambda$ = 0.154 nm).

The variation of initial of chemical composition along with the special growth conditions allowed one to obtain up to 50 mm long transparent optically perfect crystals. Two plane–parallel polished plates were made from the grown boule. In one plate, the crystallographic $c$ axis coincided with the plane of the plate, while in the second one, the $c$ axis was perpendicular to the said plate plane. The thickness of the plates ($2.225 \pm 0.005$ mm) was measured using a digital point Mitutoyo 342-251-30 micrometer ($\pm 0.002$ mm accuracy). The surface area of the obtained samples made allowed for the measuring of their optical properties using a diaphragm with a diameter of 8 mm, which is larger than the radiation wavelength. This eliminates diffraction distortion in the obtained transmission spectra over the entire experimental frequency range.

The low-frequency transmission spectra of the $LiGdF_4$ single crystal in the 0.3–3 THz (10–100 $cm^{-1}$) range were recorded using the laboratory THz time-domain spectrometer (TDS) [26]. Linearly polarized THz radiation was obtained using free-standing wire grid. The polarization degree was measured as a transmission parallel ($T_{max}$) and crossed the ($T_{min}$) grid polarizer and analyzer according to the following equation: $P = (T_{max} - T_{min})/(T_{max} + T_{min})$, and it was found to be equal to 0.99.

The Bruker IFS113v IR Fourier spectrometer was used to record transmission and reflection spectra in the 30–5000 $cm^{-1}$ range. A periodic metal grid (1200 lines/mm) on a polyethylene substrate provided linear polarization of IR radiation, with a degree of polarization of 0.975.

The spectra were recorded with the electric field vector E of the radiation parallel and perpendicular to the $c$ axis of the crystal.

Spectra simulations were performed simultaneously for all transmission and reflection data sets. The Fresnel equations were used for reflection spectra modelling, while absorption bands were calculated using the classical oscillator additive model. Three adjustable

parameters—the dielectric contribution of the $j^{-th}$ optical phonon mode $\Delta\varepsilon_j$, its frequency $v_j$ [cm$^{-1}$], and the damping constant $\gamma_j$ [cm$^{-1}$]—were determined using Equation (1), as follows:

$$\varepsilon * (v) = \varepsilon_\infty + \sum_{j=1}^{N} \frac{\Delta\varepsilon_j v_{jTO}^2}{v_{jTO}^2 - v^2 + iv\gamma_{jTO}}. \tag{1}$$

The interference pattern in the transmission spectra in the THz range was simulated using the model of coherent transmission of a plane-parallel layer [27–29].

Difference multiparticle absorption in the THz range [22,23,30,31] was simulated using the classical oscillator model. The dispersion of the transmission spectra at the high-frequency edge of phonon absorption was separately calculated using the Cauchy–Sellmeier (C-S) approach. The obtained parameters of polynomials (not listed in the tables) were used only to calculate the response function spectra at the frequencies above 1000 cm$^{-1}$.

### 3. Experimental Results

The phase composition of the LiGdF$_4$ crystal was confirmed by the X-ray diffraction method (XRD). The calculation of the lattice parameters ($a = b = 5.217(4)$, $c = 10.976(5)$ Å, tetragonal system, SSG $I4_1/a$) was carried out using HighScore Plus software.

The XRD data of the LiGdF$_4$ powdered samples (crushed single crystal) are presented in Figure 1 together with XRD standard data.

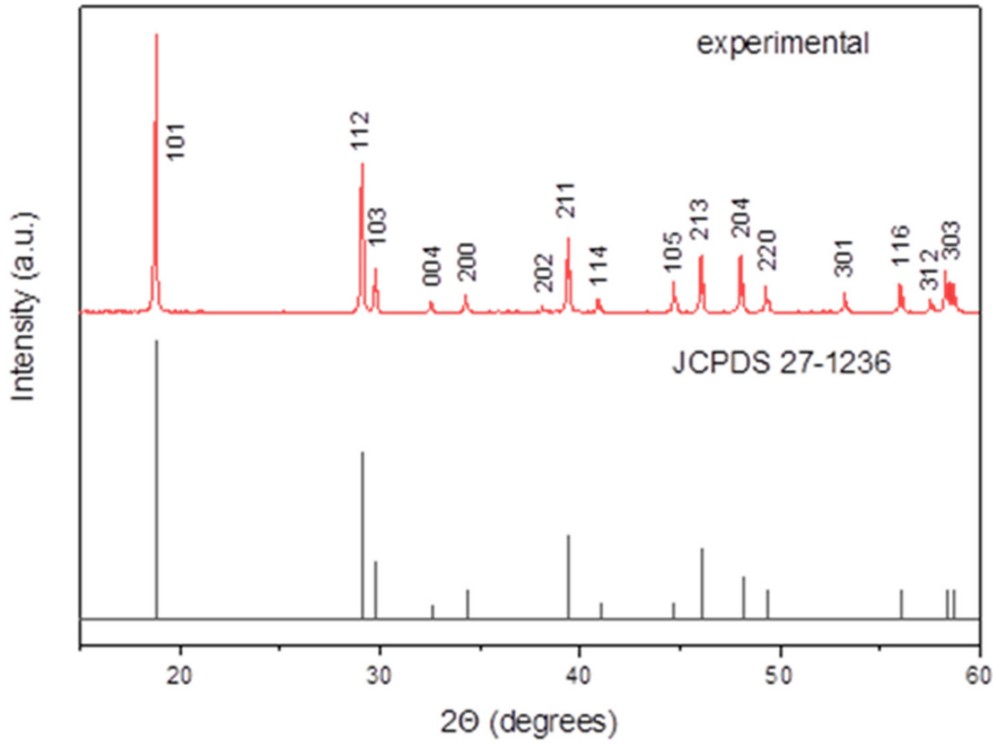

**Figure 1.** XRD pattern of the LiGdF$_4$ sample and XRD standard of LiGdF$_4$.

The LiGdF$_4$ samples are transparent in near IR and visible spectral ranges and have intensive absorption bands in UV due to 4f-4f transitions from the $^8S_{7/2}$ ground state of the Gd$^{3+}$ ions to the $^6P_j$, $^6I_j$, and $^6D_j$ excited ions, being outside of the said spectrum range (Figure 2).

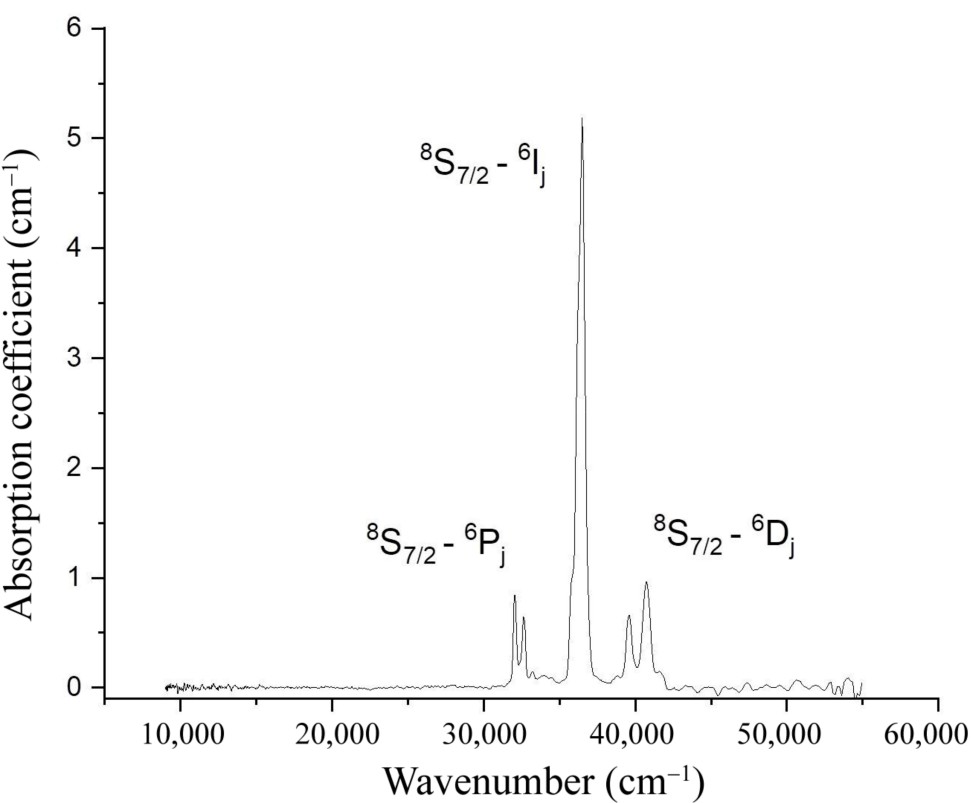

**Figure 2.** Non-polarized room-temperature absorption spectrum of LiGdF$_4$ crystal.

## 4. THz-IR Spectra Fitting and Parameters Calculation

Broadband experimental and calculated spectra of LiGdF$_4$ single crystal for E∥c and E⊥c polarization are shown in Figure 3 by dots and solid lines, respectively. The factor group analysis [32,33] predicts 4 A$_u$ optical phonon modes and 4 E$_u$ modes corresponding to the directions of E polarization parallel and perpendicular to the *c* axis. The reconstruction of the parameters of optical phonon modes was performed based on the reflection spectra and with the use of the Equation (1) under the assumption of semi-infinite flat sample. The determined parameters of optical phonon modes of LiGdF$_4$ single crystal are listed in Tables 1 and 2. Semi-infinite approximation is valid in the frequency range of high absorption. However, experimental reflection data increased due to incoherent reflection from the rear face of the sample in the transparent area above 1000 cm$^{-1}$.

The limits of applicability of fitting the reflection spectra of optical phonon modes by the additive classical oscillator model (Figure 3a,b) are illustrated by the graphs of the transmission spectra (Figure 3 c,d) in order to reconstruct the crystal response function. The transmission spectra were calculated in the coherent mode and contain interference patters in the low- and high-frequency transparency regions. The said transmission spectra are calculated using the classical oscillator model in accordance with the obtained parameters of the optical phonon modes (denoted as *Cl.Osc.*); they have an reduced value compared to the experiment in the mid-IR range. In order to obtain the absorption values in the mid-IR spectrum range, the transmission spectra were fitted using the Cauchy–Sellmeyer approach. In the THz range, the transmission spectra were calculated using the parameters of IR optical phonon modes, which are shifted to the higher frequencies compared to the experimental data. The eigenfrequency of the lowest optical phonon modes A$_u$(1) and E$_{1u}$(1) (Table 1 and 2) are 195.5 and 129.2 cm$^{-1}$, respectively, and their contributions are not sufficient to fit the experiment. One can expect both intrinsic and extrinsic absorption in the THz spectrum range. The extrinsic absorption can be neglected due to the high optical quality and low defect concentration in the grown crystals. The intrinsic absorption includes both the contribution of optical phonon modes and two-phonon difference processes [22,23].

Absorption due to different multiparticle processes in the THz range was described using Equation (1) with the substitution of large values of the damping constant $\gamma$. The parameters of the oscillators are listed in Tables 1 and 2 and denoted as $D_{mp}$. Additional absorption was also present at the frequencies of the highest reflection bands (LO-TO phonon splitting). These absorption bands are associated with second-order two-phonon processes [34], for which the photon absorption leads to the creation of two optical phonons on the Brillouin zone boundary with opposite momenta. The parameters of these absorption bands are denoted as $\sum_{mp}$.

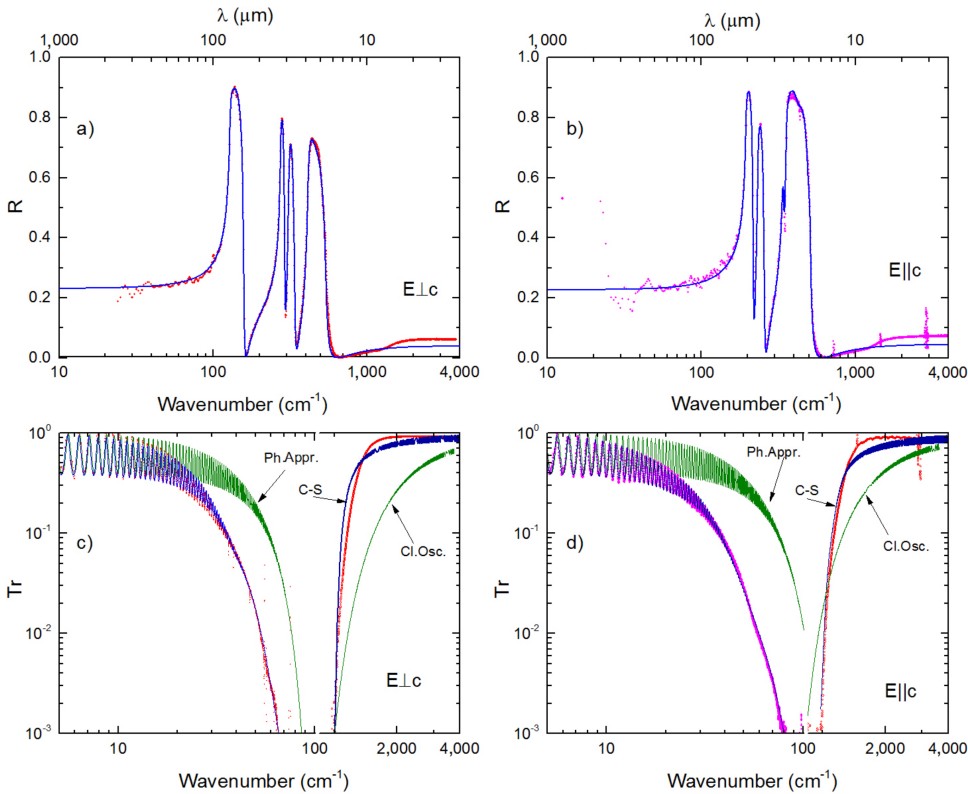

**Figure 3.** Reflection (**a**,**b**) and transmission (**c**,**d**) spectra on the LiGdF$_4$ single crystal obtained in E$\perp$c and E$\parallel$c. Dots show experimental data. Solid lines show calculated spectra. Lines marked (Ph. Appr.) show extrapolation to the THz range of the calculated spectra in the phonon approximation; lines marked *Cl.Osc.* are calculated transmission spectra using parameters of classical oscillator model; and lines denoted as *C-S* show the simulated spectra using Cauchy–Sellmeier approach.

**Table 1.** Parameters of oscillators simulating the absorption bands of the LiGdF$_4$ single crystal for E$\parallel$c. $\varepsilon_\infty$ = 2.36(7) n = 1.53(8).

| $N_{osc}$ | Assign. | $\Delta\varepsilon$ | $\nu$, cm$^{-1}$ | $\gamma$, cm$^{-1}$ | f, cm$^{-2}$ |
|---|---|---|---|---|---|
| 1 | $D_{mp}$ | 0.011 | 21.5 | 18 | 5.5 |
| 2 | $D_{mp}$ | 0.066 | 58.6 | 60 | 229 |
| 3 | $D_{mp}$ | 0.05 | 95 | 60 | 475 |
| 4 | $A_u(1)$ | 2.8 | 195.5 | 3.7 | 10,845 |
| 5 | $A_u(2)$ | 0.66 | 232.5 | 5.8 | 35,679 |
| 6 | $A_u(3)$ | 0.88 | 338.5 | 11 | 10,0846 |
| 7 | $A_u(4)$ | 1.02 | 354.8 | 9.7 | 128,983 |
| 8 | $\sum_{mp}$ | 0.005 | 494 | 37 | 11.79 |
| 9 | $\sum_{mp}$ | 0.026 | 423 | 65 | 4568 |

**Table 2.** Parameters of oscillators simulating the absorption bands of the LiGdF$_4$ single crystal for E⊥c. $\varepsilon_\infty$ = 2.25(4) n = 1.50(1).

| $N_{osc}$ | Assign. | $\Delta\varepsilon$ | $\nu$, cm$^{-1}$ | $\gamma$, cm$^{-1}$ | f, cm$^{-2}$ |
|---|---|---|---|---|---|
| 1 | $D_{mp}$ | 0.028 | 41 | 29 | 47 |
| 2 | $D_{mp}$ | 0.04 | 64 | 38 | 177 |
| 3 | $D_{mp}$ | 0.04 | 84 | 28 | 298 |
| 4 | $E_u(1)$ | 3.05 | 129.2 | 2.9 | 50,906 |
| 5 | $E_u(2)$ | 1.22 | 275.3 | 5.1 | 92,128 |
| 6 | $E_u(3)$ | 0.51 | 309.7 | 8.9 | 49,031 |
| 7 | $E_u(4)$ | 0.8 | 413.8 | 15.2 | 137,600 |
| 8 | $\Sigma_{mp}$ | 0.17 | 432 | 102 | 31,963 |

## 5. Absorption Processes in the THz Range

The spectral data obtained in the THz range for a LiGdF$_4$ single crystal with linear polarized radiation are shown in Figure 4a,b. High-resolution transmission spectra (Figure 4a) indicate the differences in the recorded interference patterns. Interference in the spectra was used to calculate the refractive indexes according model [27] for both ordinary and extraordinary beams in LiGdF$_4$ single crystal (n$_o$ = 2.85 and n$_e$ = 2.81, the birefringence value is $\Delta$n = 0.04).

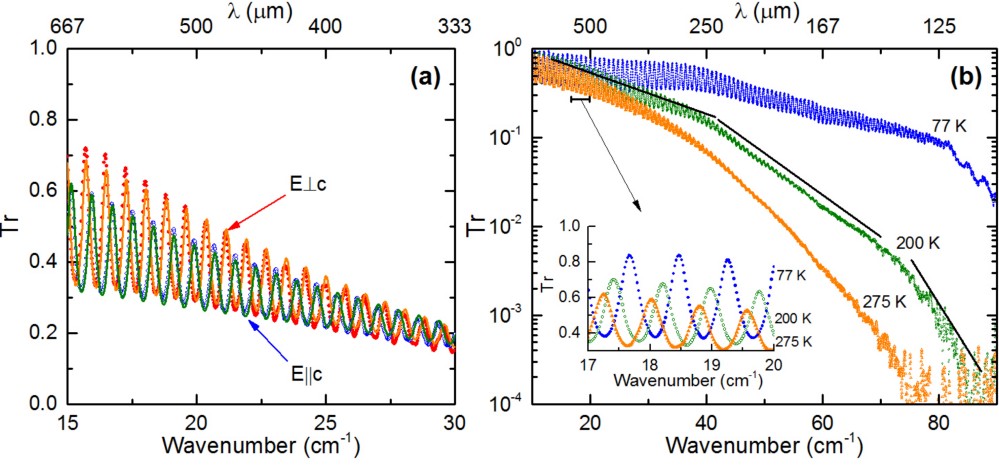

**Figure 4.** Scaled part of transmission spectra of the LiGdF$_4$ single crystal in the THz range for E⊥c and E∥c polarization. Points represent the experimental data, and solid lines correspond to the spectra calculated by Fresnel–Lorentz model (**a**). Temperature dependence of the transmission spectra of the single crystal LiGdF$_4$ in E⊥c polarization (**b**).

The deconvolution of the phonon and multiparticle absorption processes was carried out with the use of the temperature dependence of transmission spectra. The temperature evolution of transmission of LiGdF$_4$ single crystal for E⊥c polarization is shown in Figure 4b. Two processes led to the observed temperature-dependent changes: the narrowing of the phonon absorption contour and the decrease in the probability of transitions between phonon branches due to the reduction in their population [22].

The changes in the slopes of the transition spectra in different frequency regions (Figure 4b) indicate the presence of overdamped absorption processes. The total dielectric permittivity of optical phonon modes is assumed to be constant due to the fulfillment of the sum rule and the absence of structural phase transitions. In the insert Figure 4b, it can be seen that the interference period of the transmission spectra increases with cooling; this fact indicates a decrease in the refractive index and, hence, the dielectric permittivity.

## 6. Discussion

The spectra decomposition into elementary excitations was carried out using calculations by the classical oscillator model. The deconvolution of the absorption processes in the response function of a LiGdF$_4$ single crystal was carried out using the spectra of the complex dielectric permittivity. Two independent approaches were used to reconstruct the spectra of real $\varepsilon'(\nu)$ and imaginary $\varepsilon''(\nu)$ parts of the dielectric permittivity. In the first method, these spectra were calculated by fitting the parameters listed in Tables 1 and 2. The second method was to apply the Kramers–Krönig relations to the IR reflectivity spectra $R(\nu)$ to reconstruct the phase spectra of reflected waves $\varphi(\nu)$. The experimental reflectivity and reconstructed phase shift spectra were used to calculate $\varepsilon'(\nu)$ and $\varepsilon''(\nu)$. In Figure 5, the solid lines show the spectra calculated using the model parameters, and the dots show the spectra obtained using the Kramers–Krönig relations. The spectra obtained by the Kramers–Krönig method retain all the noise and distortions of the experimental reflection spectra. They also contain distortions in the low-absorption regions at frequencies between the two optical phonon modes (150–250 cm$^{-1}$ E$\perp$c, and 200–300 cm$^{-1}$ E$\parallel$c). Nevertheless, the Kramers–Krönig method provides an independent way to control the modelling reliability of the contribution of the optical phonon modes.

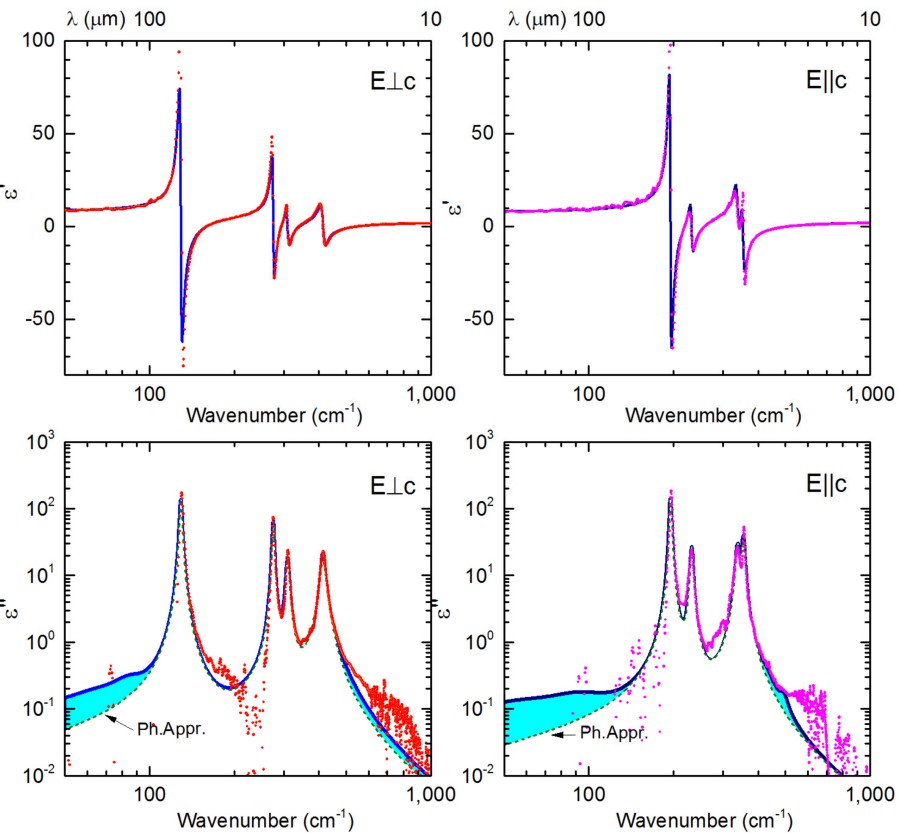

**Figure 5.** Spectra of complex dielectric permittivity of LiGdF$_4$ single crystal along principal crystallographic direction with E$\perp$c and E$\parallel$c. Solid lines show spectra simulated using parameters listed in Tables 1 and 2. Dashed lines show the extrapolation of the simulated spectra using only A$_u$ and E$_u$ phonons contribution. Dots show spectra obtained using Kramers–Krönig relations.

The classical oscillator model provides a correct shape of the resonance line near the eigenfrequencies of the optical phonon modes. The low-frequency extrapolation of the dielectric loss spectra $\varepsilon''(\nu)$ of the A$_u$ and E$_u$ phonon modes, depicted by dashed lines in Figure 5 (Ph.Appr.), indicates that their values are approximately an order of magnitude smaller than $\varepsilon''(\nu)$ spectra calculated based on transmission data (solid lines, Figure 5).

The phonon contribution to the dielectric loss was calculated based on reflection spectra; it is shown by dashed lines in Figure 6 for 298 K and 77 K temperatures. The temperature dependence of the eigenfrequency of the phonon mode follows the Grüneisen relation [35–37]. A change in the sample temperature within about 200 K range corresponds to the blue shift eigenfrequency approximately of 1–3 cm$^{-1}$. The decrease in cooling of phonon absorption in the THz range due to narrowing of the resonance contour has been determined by the damping constant according to the Bose–Einstein model [35].

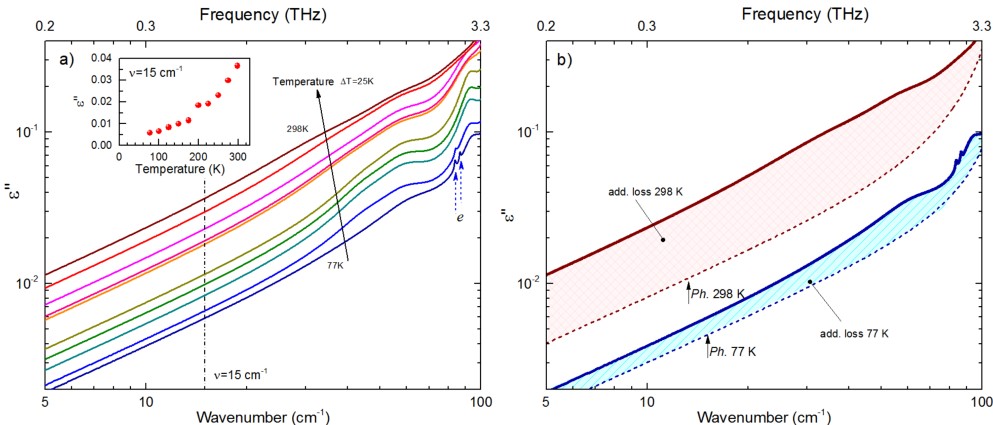

**Figure 6.** Temperature evolution of $\varepsilon''(\nu)$ spectra in the THz frequency range. Solid lines on graphs (**a**,**b**) indicate the total level of the dielectric losses, obtained using THz transmission spectra. Dashed lines on graph (**b**) indicate the level of dielectric losses determined by the contribution of optical phonon contours at temperatures of 298 and 77 K. The shaded areas show additional losses (add.loss) with respect to one-phonon absorption.

Solid lines in Figure 4 show the spectra of total losses in the THz range. The spectra were acquired with 25 K temperature steps. The wide absorption band has been revealed in the spectra $\varepsilon''(\nu)$ in the frequency range of 40–70 cm$^{-1}$ with temperature decrease. Dielectric losses exhibit non-monotonic reduction under cooling outside of the dispersion range of this band (Figure 6a insert). Such temperature behavior of the loss spectra may reflect a decrease in the population of acoustic branches under cooling and residual absorption due to transitions between optical branches.

Another type of absorption in the form of narrow resonances at 83.9 and 86.6 cm$^{-1}$ frequencies has been observed below 100 K. These bands can be associated with 5d-4f electron orbital transitions having a mixed electro and magnetic dipole character [38]. The electro dipole transition between f-states is forbidden in centrosymmetric Gd$^{+3}$ (4f$^7$5d$^1$) ion sites. However, the low symmetry of the LiGdF$_4$ crystal lattice promotes the splitting J ground state into Kramers doublets, so inter-multiplet transitions take place. Figure 6b illustrates the decrease in the contribution of multi-particle processes to the loss spectra (denoted as add.loss) under cooling from 298 K to 77 K by approximately factor of 3. In this chart, changes in the low-frequency wings of the phonon absorption contours are taken into account and are shown by dashed lines.

Thus, in the THz range, three main intrinsic absorption processes represent the electrodynamic response of the LiGdF$_4$ crystal: the low-frequency wing of the optical phonon mode, the multipartile difference processes, and the contribution of the electronic transition between the split multiplet levels of the Gd$^{+3}$ ion ground state.

Figure 7 shows the calculated spectra of the refractive index and the absorption coefficient of the LiGdF$_4$ single crystal in the THz-IR ranges at room temperature, which are of practical interest for the non-linear optical applications.

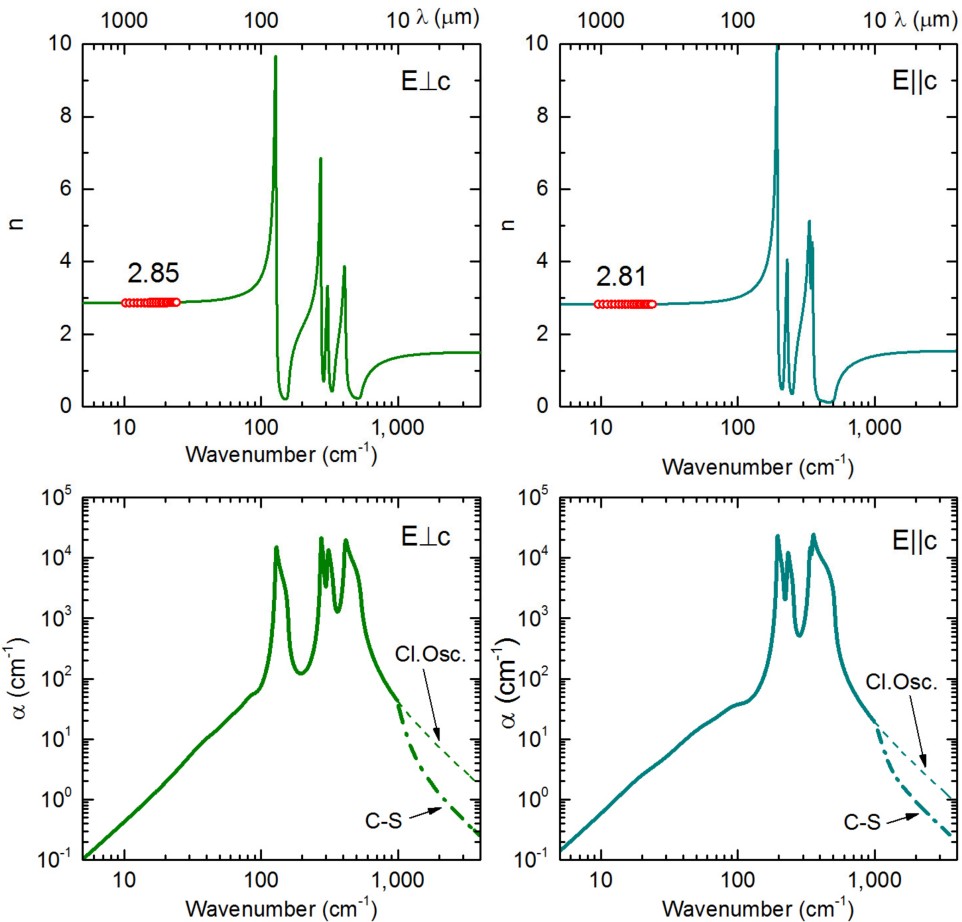

**Figure 7.** Calculated spectra of the refractive index and absorption coefficient of the LiGdF$_4$ single crystal in the THz-IR ranges at room temperature. Solid lines are the spectra n($\nu$) and $\alpha$($\nu$) calculated using parameters given in Tables 1 and 2, and hollow circles are refractive index values obtained directly from interference pattern using model [27].

Refractive index values, calculated using fitting parameters (solid lines) and obtained using equations of the coherent transmittance of single-layer sample (hollow circles), are presented in the said Figure 7 for the THz range. In the mid-IR range, absorption spectra, obtained based on the classical oscillator model (dashed lines Cl.Osc.), were corrected using Cauchi–Sellmeyer equations (dash-dotted lines C-S). Thus, the optical parameters of the LiGdF$_4$ single crystal in the THz-IR ranges were restored with sufficient accuracy.

## 7. Conclusions

LiGdF$_4$ single crystals were grown by Bridgman–Stockbarger technique. The dimensions of prepared single crystals were suitable for optical measurements in a wide frequency range without diffraction distortions. The decoding of the electric-dipole absorption spectrum in a crystal into elementary excitations has been carried out using the model of a classical oscillator. It allowed one to determine the parameters of the optical phonon modes of the LiGdF$_4$ crystal in the main crystallographic directions and to highlight the contribution of multiparticle difference processes that increased absorption in the THz frequency range by about an order of magnitude.

The optical characteristics of the crystal were established using direct measurements of transmission spectra with high-frequency resolution. The birefringence in the THz range $\Delta n = 0.04$ was calculated. The dependence of the absorption coefficient of the crystal on temperature in the THz range, determined by both a single-phonon contribution and by multiparticle processes, has been found. The dispersion of the crystal absorption coefficient

at the high-frequency edge of phonon absorption has been estimated using the Cauchy–Sellmeyer approach. It is worth noting that, in the THz range, the behavior of the response functions of a rare-earth fluoride crystal and oxides has a qualitative similarity. The main mechanisms, determining the absorption and dispersion of the optical parameters of this crystal in the THz and IR ranges, were determined and analyzed.

**Author Contributions:** G.A.K.: Conceptualization, methodology, formal analysis, investigation, writing—original draft preparation, writing— review and editing, and visualization. S.P.L.: Methodology, formal analysis, investigation, software, and writing—review and editing. S.L.K.: Crystal growth and sample preparation. O.A.M.: Crystal growth and sample preparation. V.M.K.: Investigation. V.V.S.: Writing—review and editing. P.P.F.: Conceptualization, writing—review and editing. All authors have read and agreed to the published version of the manuscript.

**Funding:** This work was carried under the R&D plan of the Prokhorov General Physics Institute of the Russian Academy of Sciences.

**Data Availability Statement:** The data that support the findings of this study are available within the article.

**Acknowledgments:** The crystal growth experiments and primary samples' characterization were carried out in accordance with the Strategic Academic Leadership Program "Priority 2030" of the Kazan Federal University of the Government of the Russian Federation and were funded by the government assignment for FRC Kazan Scientific Center of RAS. The authors express their gratitude to A. R. Hadiev for measuring the absorption spectra of $LiGdF_4$ crystals in the near infrared, visible, and UV ranges. The authors express their sincere gratitude to Arthur I. Popov for his most kind assistance in the preparation of the present manuscript.

**Conflicts of Interest:** The authors declare no conflict of interest.

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
