# Peer review of "Optical Properties of LiGdF4 Single Crystal in the Terahertz and Infrared Ranges"

_photonics, doi:10.3390/photonics10010084_

Round 1

Reviewer 1 Report

The authors present a detailed characterization of temperature-dependent optical properties of LiGdF4 single crystal performed by careful measurements in the THz and IR frequency ranges for two principal polarizations. They determine and identify the absorption mechanisms and model them with appropriate dispersion models. The quantitative data presented will definitely be helpful in designing devices based on the compound and is thus worth publishing. However, the presentation of the nice results can be significantly improved. I recommend the authors to revise the manuscript with respect to the language used and suggest them ask a native English speaker or equivalent to assist with correcting the spelling, grammar, word use, and punctuation throughout the manuscript. Once corresponding procedures are done, I would definitely recommend the manuscript to be published in Photonics.

Author Response

We really appreciate the reviewer’s and editor’s thorough reading our manuscript and their suggestions that improved our paper. We have made all the suggested text corrections.

The text of the article was checked and corrected by a native English speaker (American version).

Reviewer 2 Report

This work by Komandin et al. presents broadband polarized IR and THz reflection and absorption spectra of single crystal LiGdF4. The data is analyzed following the harmonic oscillator model to extract the parameters of the optical phonon modes, and the contribution of multiparticle absorption processes is discussed.

The used methods (THz Time-Domain spectroscopy, Fourier-Transform Infrared spectroscopy) are appropriate tools to determine the optical parameters of the high-purity LiGdF4 crystal, and the analysis and discussion is sound.

I suggest the authors make some revisions prior to publication.

1) The order of first name, middle initial, and last name should be checked to match the style of the journal and to avoid confusion.

2) p.2, l.88: “THz pulsed spectrometer (TPS)” is a rather uncommon name for what is usually known as a “THz time-domain spectrometer”. A description of the setup used in this study, as given in the cited paper [26], also uses the term “THz time-domain spectrometer”. I advise to use the term “THz time-domain spectrometer” for consistency and to avoid possible confusion for readers.

3) In Fig. 1 both experimental and simulated XRD spectra are shown. However, the lower panel of this figure is titled “JCPDS 27-1236” which I think is a commercial database for XRD spectra? There is a peak at ~53 deg. in the experimental data which is not visible in the simulated spectrum. What would be the explanation for this difference?

4) p.4 l.127: “Figure 1” is referenced in the text, but the authors discuss Figure 3.

5) p.5 Table 1/2.: Are these parameters a result of a “simulation” or rather the best fit parameters of the fitted spectra? Is it possible to give the standard errors of these parameters?

6) p.9 Fig. 7 Just following the figure caption it is not entirely clear what the red circles are. In the text it is only stated “values of the refractive index […] obtained using equations of the coherent transmittance of single-layer sample”. Could you please clarify/elaborate on this? Also, it should be possible to access the refractive index directly from the THz Time-Domain data? This would also help with Fig. 4b (inset), where the interference period is discussed with respect to the refractive index.

7) While the content of the manuscript can mostly be understood without greater difficulty, the language used in the text needs to be revised. There may be issues with grammar, word choice, or other elements of language usage that could be improved upon to make the text clearer.
Examples: p.2 l.89 “Linearly polarized THz radiation obtained using free-standing wire grid.” (missing word) or p.2 l.98 “Spectra simulating were performed simultaneously for all transmission and reflection data set” (grammar), or p.7 l.203 “Kramers-Krönig” (spelling).
By revising the language of the manuscript, it is certain that the overall quality and readability of the text will be significantly enhanced.

Author Response

We really appreciate the reviewer’s and editor’s thorough reading our manuscript and their suggestions that improved our paper. We have made all the suggested text corrections.

1) The order of first name, middle initial, and last name should be checked to match the style of the journal and to avoid confusion.

 We have made corrections in accordance with the Photonics-template

2) p.2, l.88: “THz pulsed spectrometer (TPS)” is a rather uncommon name for what is usually known as a “THz time-domain spectrometer”. A description of the setup used in this study, as given in the cited paper [26], also uses the term “THz time-domain spectrometer”. I advise to use the term “THz time-domain spectrometer” for consistency and to avoid possible confusion for readers.

We have checked and corrected this issue.

3) In Fig. 1 both experimental and simulated XRD spectra are shown. However, the lower panel of this figure is titled “JCPDS 27-1236” which I think is a commercial database for XRD spectra? There is a peak at ~53 deg. in the experimental data which is not visible in the simulated spectrum. What would be the explanation for this difference?

We have corrected an inaccuracy in the caption to the Fig. 1. This figure really shows an experimental X-ray pattern of LiGdF4 in comparison with the reference data from the Joint Committee on Powder Diffraction Standards (JCPDS) commercial database. The presence of the latter information in the figure unequivocally demonstrates that our data are more complete.

4) p.4 l.127: “Figure 1” is referenced in the text, but the authors discuss Figure 3.

We have checked and corrected this issue.

5) p.5 Table 1/2.: Are these parameters a result of a “simulation” or rather the best fit parameters of the fitted spectra? Is it possible to give the standard errors of these parameters?

It is difficult to determine the errors for the all listed parameters at once. We used both transmission and reflection spectra for determination of the oscillator parameters. It is known that transmission spectra are more sensitive to weak absorption lines and allow more accurate calculate their parameters. Intense lines in the spectra with large values of dielectric contribution are associated with optical phonon modes, and their parameters were determined from less accurate reflection spectra. In addition, the accuracy of spectra acquisition in different parts of the wide spectral range is also different due to the configuration peculiarities of the optical scheme, the source intensity, and the detector sensitivity. Therefore, taking into account the above said, the significant digits listed in Tables 1 and 2 may be considered as the accuracy of determining the parameters of absorption lines.

6) p.9 Fig. 7 Just following the figure caption it is not entirely clear what the red circles are. In the text it is only stated “values of the refractive index […] obtained using equations of the coherent transmittance of single-layer sample”. Could you please clarify/elaborate on this? Also, it should be possible to access the refractive index directly from the THz Time-Domain data? This would also help with Fig. 4b (inset), where the interference period is discussed with respect to the refractive index.

We corrected the Figure 7 caption and provided assignment to the solid lines and red hollow circles. Determining the complex refractive index from the interference in the transmission spectrum is the standard approach. Interference in a plane-parallel layer with absorption was used to determine the Fabry-Perot effect. The case of small dispersion of real part of refractive index can be described by the following equation: n=mc/(2d[Vmax]), where m is the order of the interference maximum, с is speed of light in free space, d is sample thickness, and [Vmax] is the frequency of the interference maximum. If the interference maxima are equidistant, a simplified expression can be used for calculation of the refractive index by n=c/(2d<delta>V) equation, where <delta>V is frequency difference between two adjacent maxima. Since this method is described in textbooks on optics (for example Max Born, Emil Volf. Principles of Optics) and well known, we considered it was possible to confine ourselves to the reference [27] with a detailed and rigorous description of the method. Please note that the standard commercial TDS software package allows calculating the complex refractive index using the changes in position and amplitude of photocurrent pulses in the reference and sample signals. However, this method leads to the artifacts in the form of oscillations in the transparency parts in the absorption spectra due to the Fabry-Perot effect. These artifacts are not so harmless for subsequent analysis, since the peaks in the absorption spectra should be related to resonant absorption mechanisms that are absent in the real material.

7) While the content of the manuscript can mostly be understood without greater difficulty, the language used in the text needs to be revised. There may be issues with grammar, word choice, or other elements of language usage that could be improved upon to make the text clearer.
Examples: p.2 l.89 “Linearly polarized THz radiation obtained using free-standing wire grid.” (missing word) or p.2 l.98 “Spectra simulating were performed simultaneously for all transmission and reflection data set” (grammar), or p.7 l.203 “Kramers - Krönig” (spelling).
By revising the language of the manuscript, it is certain that the overall quality and readability of the text will be significantly enhanced
.

The text of the article was checked and corrected by a native English speaker (American version).